# hBN Flake Embedded Al_2_O_3_ Thin Film for Flexible Moisture Barrier

**DOI:** 10.3390/ma14237373

**Published:** 2021-12-01

**Authors:** Wonseok Jang, Seunghun Han, Taejun Gu, Heeyeop Chae, Dongmok Whang

**Affiliations:** 1School of Advanced Materials Science and Engineering, Sungkyunkwan University (SKKU), 2066 Seobu-ro, Jangan-gu, Suwon-si 16419, Korea; wid0129@skku.edu (W.J.); gtj1008@skku.edu (T.G.); 2School of Chemical Engineering, Sungkyunkwan University (SKKU), 2066 Seobu-ro, Jangan-gu, Suwon-si 16419, Korea; hsh12040@skku.edu; 3SKKU Advanced Institute of Nano Technology (SAINT), Sungkyunkwan University (SKKU), 2066 Seobu-ro, Jangan-gu, Suwon-si 16419, Korea

**Keywords:** hexagonal boron nitride, exfoliation, plasma enhanced atomic layer deposition, point defect, flexible moisture barrier

## Abstract

Due to the vulnerability of organic optoelectronic devices to moisture and oxygen, thin-film moisture barriers have played a critical role in improving the lifetime of the devices. Here, we propose a hexagonal boron nitride (hBN) embedded Al_2_O_3_ thin film as a flexible moisture barrier. After layer-by-layer (LBL) staking of polymer and hBN flake composite layer, Al_2_O_3_ was deposited on the nano-laminate template by spatial plasma atomic layer deposition (PEALD). Because the hBN flakes in Al_2_O_3_ thin film increase the diffusion path of moisture, the composite layer has a low water vapor transmission ratio (WVTR) value of 1.8 × 10^−4^ g/m^2^ day. Furthermore, as embedded hBN flakes restrict crack propagation, the composite film exhibits high mechanical stability in repeated 3 mm bending radius fatigue tests.

## 1. Introduction

Optoelectronic devices using organic semiconductors have been extensively developed as mechanically flexible and solution-processable devices [1]. Especially, organic semiconductors have been used in various applications such as organic light-emitting diodes (OLEDs) [2] and organic photovoltaics (OPVs) [3] by controlling the molecular structure. Since the organic devices deteriorate rapidly when directly exposed to oxygen or moisture, it is necessary to ensure reliability by encapsulating the organic devices with a protective barrier layer that minimizes the permeation of the deteriorating gases, especially moisture [4,5]. With rigid optoelectronic devices, inorganic glass sheet has been widely used because glass is an excellent gas barrier and also easy to handle. For flexible devices, however, an alternative flexible thin-film gas barrier is required because the inorganic glass is vulnerable to mechanical stress caused by bending [6,7].

Thin-film barriers could be divided into single-layer and multi-layer thin-film barriers. In the single-layer thin film layer, inorganic materials such as Al_2_O_3_ and SiN_x_ are deposited by atomic layer deposition (ALD), chemical vapor deposition (CVD), etc. The single-layer thin film barrier could be formed by a relatively straightforward process. However, it is vulnerable to gas permeation because of pinholes and internal defects in the layer [8]. The defects in the barrier films, causing penetrating water molecules, are divided into two major types: physical defects, such as pinholes or cracks, and chemical defects formed by un-reacted radicals, low deposition temperature, and the initial state of the ALD deposition process [9,10]. As the thickness is increased to reduce defects, it would be cracked easily due to mechanical deformation [11]. As multi-layer thin-film barriers that overcome the drawback of single-layer barriers, nano-laminate structures, in which thin inorganic layers and polymers are repeatedly stacked, have been generally used. Within the nano-laminated multi-layer structures, moisture moves horizontally within the layer until it passes vertically through the defect or pinhole, increasing the diffusion path, thus enabling a relatively low water vapor transmission ratio (WVTR) value. Also, it is bendable without mechanical damage because of mechanical stress relaxation by the polymer of each layer [12]. However, the fabrication process of the multi-layer barrier film is quite complicated because each layer is deposited separately. 

As an alternative to the thin-film barriers, two-dimensional (2D) materials such as graphene and hexagonal boron nitride (hBN) have been studied as flexible gas barriers because of their excellent gas-impermeability, chemical stability, and mechanical flexibility [13,14]. In particular, hBN has great potential as a material for flexible gas barrier films because of its high optical transparency. The 2D materials have been synthesized via the CVD process using metal catalysts such as Cu [15], Ni [16], Au [17] and transferred to other substrates for use as barrier materials. However, it has been reported that the transferred 2D materials have a high WVTR value of about 10^−1^ g/m^2^ day due to mechanical defects such as tears and wrinkles occurring in the transfer process [14,17]. Depositing Al_2_O_3_ on graphene prevented moisture permeation through the transfer defects, but the light transmittance was lowered due to the light absorption of graphene [18]. Instead of using the CVD-grown 2D materials, thin-film barriers can also be fabricated by stacking 2D flakes exfoliated from the bulk 2D material with sonication [19], ball milling [20], etc. For example, the layer-by-layer (LBL) coating by electrostatic interaction between polymers and 2D flakes enables a nano-laminate structure that maximizes the diffusion path of water molecules. Nevertheless, the nano-laminate structure also allows water permeation between 2D material flakes. By filling the gaps between the 2D flakes with inorganic particles to prevent moisture permeation between the flake, a reasonably low WVTR value of ~10^−3^ g/m^2^ day was obtained [21], which still has not reached the value of 10^−4^~10^−6^ g/m^2^ day required for OLED and perovskite solar cells [22].

In this study, defect minimized and diffusion path maximized moisture barrier have been developed by single-layered hBN embedded Al_2_O_3_ film. The nano-layered structure was formed through LBL coating using hBN flakes and polyethyleneimine (PEI). Al_2_O_3_ deposited via PEALD binds to the amine group at the edge of hBN flakes and fills the gaps between the flakes. In addition, the hBN flakes prevent moisture from penetrating in the vertical direction by increasing the diffusion path. The resulting WVTR values significantly decreased due to the synergistic effect of minimizing defects and increasing diffusion pathways. Furthermore, hBN flakes inhibited crack propagation in barrier film. Based on the hBN embedded inorganic layer structure, we developed the flexible moisture barrier, which has a WVTR value of 1.81 × 10^-4^ g/m^2^ day and is stable in the fatigue bending test at a 3 mm bending radius.

## 2. Materials and Methods

### 2.1. The hBN Flake Exfoliation

The hBN powder (Sigma Aldrich, St. Louis, MO, USA) and urea (98%, Daejung Chemicals, Siheung, Gyeonggi, Korea) were mixed in a steel container at a weight ratio of 1:100 (hBN 50 mg, urea 5 g), and the mixture was milled with a rotational speed of 700 rpm for 20 h in Ar atmosphere using a planetary ball mill (Pulverisette 7, Fritsch, Idar-Oberstein, RP, Germany). To remove excess urea and disperse it in water at a concentration of 1 mg/ml, a mixed powder of urea and hBN flakes was dissolved in 50 ml of deionized water and dialyzed for 1 week using a dialysis kit (Sigma Aldrich, St. Louis, MO, USA).

### 2.2. Fabrication of LBL Template & Al_2_O_3_ Composite

First, a polyethylene naphthalate (PEN) substrate was treated with O_2_ plasma for 10 min under 100 W and 100 sccm of O_2_ gas flow (30 torr) to induce negative charges on the substrate by functionalizing -OH or -O groups on the surface [23]. After that, to adsorb positively charged PEI polymer (MW: 25,000, Sigma Aldrich, St. Louis, MO, USA) by electrostatic interaction, the negatively charged PEN substrate was dipped in a 2 wt % aqueous solution of PEI (pH 10, zeta potential of 59.3 mV) for 10 min and then rinsed with deionized water (DI) for 30s. The PEI-coated substrate was dipped in the aqueous dispersion of hBN flakes (1 mg/mL) for 10 min to adsorb the hBN flake, which has a zeta potential of −39.8 mV, and the resulting substrate was rinsed with DI for 30 s. The zeta potentials of the solutions of PEI polymer and hBN flakes are shown in Appendix A. Then, Al_2_O_3_ was deposited on the hBN/PEI substrate by spatial ALD. Al_2_O_3_ was deposited using trimethylaluminum (TMA, Sigma Aldrich, St. Louis, MO, USA) as the aluminum precursor and N_2_O as an oxygen radical source at 80 °C with a remoted plasma power of 150 W. TMA flowed with Ar gas as a carrier gas in 50 sccm and N_2_O gas in 20 sccm. To control the layer thickness, about 10 cycles per 1 nm thickness were repeated, and the moving speed of the spatial ALD was 125 mm/s.

### 2.3. Characterizations

The topology of exfoliated hBN flake was analyzed using a non-contact mode atomic force microscopy (AFM, Park System, Suwon, Gyeonggi, Korea). The thickness and morphology of LBL-Al_2_O_3_ and Al_2_O_3_ were measured by a field-effect scanning electron microscopy (FESEM, JSM-7401F, JEOL). Functional groups and defects in LBL-Al_2_O_3_ and Al_2_O_3_ were evaluated using X-ray photoelectron spectroscopy (XPS, ESCALAB 250Xi, thermo fisher scientific, Waltham, MA, USA) with an angle-resolved mode at 60° tilting. Although the accurate penetration depth of X-ray during the analysis is unknown, a signal at the top surface with less than 10 nm thickness was observed [24]. Optical transmittance of barrier film was measured by UV-vis spectroscopy (Agilent Technologies, Santa Clara, CA, USA). The WVTR was measured by the electrical Ca test. The device for Ca test was fabricated with an Al electrode, and a 300 nm thick Ca layer was deposited by thermal evaporation. The deposited Al_2_O_3_ on PEN was attached to the Ca test device with a UV-cured epoxy resin (TB3124L, 3 M). The electrical resistance of the device was recorded in real-time through a digital multimeter/switch system (Model 2750, Keithley, TEKTRONIX, Oregon, OR, USA). The WVTR values were calculated using the following equation [25]: $$\mathrm{WVTR}\;\left[g/m^{2}\;\mathrm{day}\right]=\frac{\mathrm{Total}\;\mathrm{grams}\;\mathrm{of}\;H_{2}O}{\left(l\times w\right)}\times \frac{10,000\;cm^{2}}{m^{2}}\times \frac{24\;h}{\mathrm{day}}\times \frac{1}{t}$$
where *l* and *w* are the length and width of the deposited Ca layer, respectively, and *t* is the time until the electrical conductivity becomes zero. To accelerate the measurement, all samples were measured at 85 °C and 85% relative humidity (RH), and the WVTR value was converted by multiplying by a conversion factor of 240. The conversion factor was determined by comparing the WVTR values measured at 25 °C/50% RH (25 °C and relative humidity of 50%) with those measured at 85 °C/85% RH in our previous work [25]. The bending property was measured by the WVTR before and after bending at a 3 cm bending radius 5000 times.

## 3. Results & Discussion

In order to form a uniform and defect-free layer of the hBN embedded Al_2_O_3_ composite (Al_2_O_3_/hBN/PEI) structure, the width of the hBN flake should be minimized. The hBN flakes exfoliated by the ball milling process had a very narrow width (<50 nm) and an atomically thin thickness (~2 nm) Appendix A. In the XPS spectra of the exfoliated hBN flake, NH_2_ groups (~398.8 eV) [26] and B-O groups (~191.7 eV) at the edge of the hBN flake [27] were observed (Appendix A). By LBL coating by electrostatic interaction between the hBN flakes and a positively charged PEI polymer, a ~5 nm thick layer was formed (Appendix A), and then the resulting nano-laminated film was used as a template to fill the free volume between the flakes by PEALD. The PEALD process has the advantage of being able to use various substrates at a low temperature. However, when applied energy is lowered, point defects of deposited materials are increased [10]. Figure 1 shows the deposition process of Al_2_O_3_ to form hBN embedded Al_2_O_3_ structure. On a bare PEN substrate, a uniform Al_2_O_3_ layer is obtained with a thickness of 1 nm per 10 ALD cycles. When Al_2_O_3_ is deposited on the hBN/PEI template, however, Al_2_O_3_ is nucleated in the voids of the hBN/PEI template and fills the void before being deposited as a uniform layer (Figure 1A). The thickness of the hBN/PEI templated Al_2_O_3_ layers both at the initial state (50 cycles of ALD deposition), and the intermediate state (100 cycles of ALD deposition) of Al_2_O_3_ deposition on the hBN/PEI template was estimated to be about 5–8 nm by measuring the thickness at more than 20 different points that are randomly selected from 5 different SEM images. At the initial (Figure 1B) and intermediate states (Appendix A), the thickness of the composite layer does not appear to change because Al_2_O_3_ mostly nucleates at the edges of the hBN flakes and then deposits in the free voids between the hBN flakes. At the saturation state of Al_2_O_3_ deposition on hBN/PEI (200 ALD cycles), the Al_2_O_3_ is further deposited on the top surface of the composite after filling the free voids between the hBN flakes (Figure 1C), resulting in the formation of 20 nm thick composite layer. The thickness of the composited layer at the saturation state was measured with cross-sectional TEM images (Figure 1D).

The nucleation mechanism of the deposition process was investigated by analyzing the chemical bonding and defects of each state by XPS analysis (Figure 2). In the early state of depositing Al_2_O_3_ with a thickness of 0–5 nm on the nano-laminated hBN/PEI layer, peaks (~77.1 eV for Al 2p and ~534.4 eV for O 1s) corresponding to aluminum oxynitride (Al-O-N) was predominantly observed [28]. Peaks related to BO_x_ (~192 eV) [27] and NO_2_ bonds (~403.5 eV) [29] were also identified, suggesting that covalent bonds were formed between the amines at the hBN edge and Al_2_O_3_ (Appendix A). Also, in the N 1s spectra, peaks corresponding to NH_3_^+^ (401.3 eV) [26] and –NH– (400.1 eV) [30] in PEI were observed. In the intermediate state of depositing 5~10 nm thick Al_2_O_3_, the intensities of the peaks corresponding to Al-OH (75.6 eV) increased, indicating that the defects of Al_2_O_3_ increased as the free volume between hBN flakes was partially filled with Al_2_O_3_ after nucleation [31]. In the saturation state of depositing Al_2_O_3_ with the thickness of >10 nm, Al-OH bonds are converted to Al-O bonds due to reaction with oxygen radicals as the number of deposition cycles increases [10,32].

In the XPS spectra of Al_2_O_3_ deposited on a bare PEN substrate, the Al-OH peaks related to the point defect gradually decrease as the thickness increases, but it still remains that the thickness increased to 20 nm (Appendix A). In the 20 nm Al_2_O_3_ deposited in the hBN/PEI template, however, the XPS peak related to point defects almost disappeared because Al-OH bond was reduced by the formation of aluminum oxynitride in the early nucleation stage_._ In addition, after Al_2_O_3_ deposition in hBN/PEI template and bare PEN, the optical transparency was more than 95%, confirming that it could be used for optoelectronic devices such as solar cells and OLED devices (Appendix A).

To confirm the moisture and ion permeation through defects of the deposited Al_2_O_3_, Cu was electrochemically deposited on the Al_2_O_3_ and LBL Al_2_O_3_/Au film. The electro-deposition was performed at −0.3 V, which is below the break-down voltage of dielectric Al_2_O_3_ film. Since the diameter of Cu^2+^ ion (0.73 Å) is smaller than that of the water molecule (2.75 Å), the absence of diffusion path of Cu^2+^ ion indicates excellent gas barrier performance of the composite. As shown in the inset scheme of Figure 3, Cu^2+^ ions penetrate along Al_2_O_3_ defects or pinholes to Au film, and Cu particles are nucleated. The tendency of penetrated Cu^2+^ was confirmed by the formation of Cu particles with a diameter larger than 1 μm. In 5 nm thick Al_2_O_3_ deposited on the bare substrate, Cu^2+^ ions rapidly penetrated through cracks or point defects of the Al_2_O_3_ layer and nucleated to form Cu particles in high density of ~10^5^ cm^−2^ (Figure 3A–C). As the thickness increased to 20 nm, the Cu particle density decreased due to the reduction of point defects, but the density was still ~10^3^ cm^−2^. However, when Al_2_O_3_ is deposited on the hBN/PEI template, the Cu particle density is significantly lower than the value of Al_2_O_3_ on a bare PEN substrate (Figure 3D,F). Especially, when 20 nm thick Al_2_O_3_ was deposited on hBN/PEI template, the particle density was minimized to less than 50 cm^−2^. This decrease in ion permeability can be attributed to (i) the decrease in the defect density of Al_2_O_3_ and (ii) the increase in the diffusion path through the hBN flake. The defect density of Al_2_O_3_ can be sufficiently decreased when Al_2_O_3_ is deposited on a bare polymer substrate with thickness over 30 nm [32]. When Al_2_O_3_ is deposited on the hBN/PEI template, however, the point defect density is significantly low even at the thickness less than 10 nm as nucleation occurs in the amine-group of exfoliated hBN. Also, even if the density of point defects is similar, Cu^2+^ ions permeate remarkably slowly because they do not penetrate through the vertical direction of hBN flakes despite Al_2_O_3_ defects.

The WVTR value of each thin film was measured using an electrical Ca test (Appendix A). Figure 4A shows that Al_2_O_3_ deposited on the hBN/PEI template was about 10 times lower than when deposited on the bare substrate. In particular, the WVTR value was significantly improved to 4.7 × 10^−4^ g/m^2^ day at a thickness of 5 nm, but the space between the hBN flakes was not completely filled, so moisture permeated through the gap. After Al_2_O_3_ was deposited to 10 nm (intermediate state), the WVTR value was gradually decreased to 4.0 × 10^−4^ g/m^2^ day. In the saturated state where Al_2_O_3_ was deposited with a thickness of 20 nm, the WVTR value was significantly reduced to 1.8 × 10^−4^ g/m^2^ day because Al_2_O_3_ completely filled the void space between hBN flakes, and point defects were significantly reduced. When 10 nm thick Al_2_O_3_ was additionally deposited after the saturation, the WVTR value was only slightly reduced to 1.5 × 10^−4^ g/m^2^ day.

We note that the WVTR of 1.8 × 10^−^^4^ g/m^2^ day at 20 nm is a significantly lower value than those of previously reported single-layer thin film barriers (Figure 4B) [17,33,34,35,36,37,38,39,40]. While bare ALD and PEALD have high WVTR values at thin thickness due to point defects and pinholes at thin thickness, in the case of Al_2_O_3_/hBN/PEI composite, hBN flake and Al_2_O_3_ in the composite form a bond and Al-OH formation is suppressed. As a result, the point defects were reduced, the diffusion path of moisture increased, and the WVTR value was significantly reduced. In addition, the flexibility was confirmed by measuring the WVTR value after 5000 cycles of bending fatigue test at a bending radius of 3 mm, which corresponds to 4% stain applied (Figure 4C) [41]. Even after the bending test, the WVTR value increased to less than 30%. As illustrated in the inset of Figure 4C, the mechanical stress can be released due to the PEI polymer layer, and crack propagation can also be restricted by hBN flakes in the composite [42,43]. For Al_2_O_3_ layers deposited on bare PEN substrate, however, WVTR values increased by more than 200% after the bending test (Appendix A).

## 4. Conclusions

In this study, a flexible single-layered gas barrier film with minimal point defects was developed using hBN embedded Al_2_O_3_ film with thickness less than 20 nm. The hBN/PEI template for nano-laminate structure was deposited by LBL method and Al_2_O_3_ was deposited by PEALD. The hBN flakes minimized point defects of deposited Al_2_O_3_ and maximized the diffusion path of water vapor, resulting in a low WVTR value of 1.81 × 10^−4^ g/m^2^ day even with 20 nm thickness. We investigated the mechanism of defect-free Al_2_O_3_ deposition and WVTR improvement by confirming the hBN edge nucleation and defect density of Al_2_O_3_ through XPS spectra, Cu electrochemical deposition, and electrical WVTR measurements. In addition, improvement of mechanical stability by hBN flakes was confirmed by measuring WVTR values before and after the bending fatigue test. The embedding flexible and transparent 2D layers in the ultra-thin inorganic barrier layer may provide a practical approach for an efficient gas barrier for flexible organic optoelectronic devices. In this study, a WVTR value of about 10^−4^ g/m^2^ day was obtained even with a single layer with a thin thickness through the hBN embedded Al_2_O_3_ structure. To apply this approach to OLED applications, it is necessary to further research for WVTR improvement through multi-layer staking.

## Figures and Tables

**Figure 1 materials-14-07373-f001:**
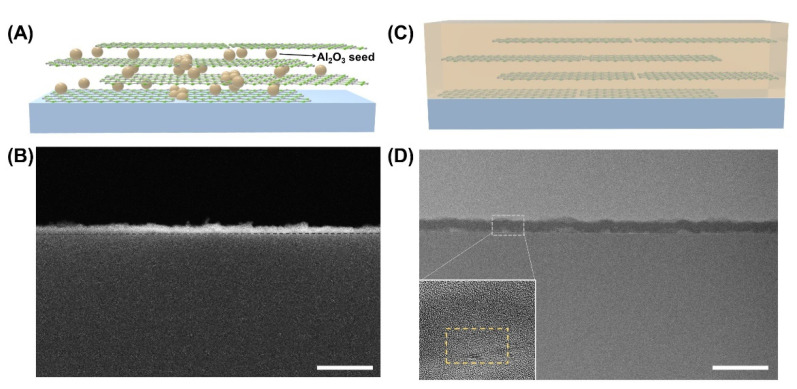
hBN templated Al_2_O_3_ deposition process. (**A**) Scheme illustrating initial state of Al_2_O_3_ deposition in hBN/PEI template layer, (**B**) A cross-sectional SEM image of Al_2_O_3_/hBN/PEI composite layer after 50 cycles ALD deposition in hBN/PEI template layer. The dashed line in (**B**) indicates the interface between the PEN substrate and the composite layer. (**C**) Scheme illustrating saturation state of Al_2_O_3_ deposition in hBN/PEI template layer after 200 cycles ALD deposition in hBN/PEI template layer, (**D**) Cross-sectional TEM images of Al_2_O_3_/hBN/PEI composite layer. The inset image is a high-resolution TEM image showing hBN flakes (dashed yellow box) in the composite layer. Scale bars are 100 nm.

**Figure 2 materials-14-07373-f002:**
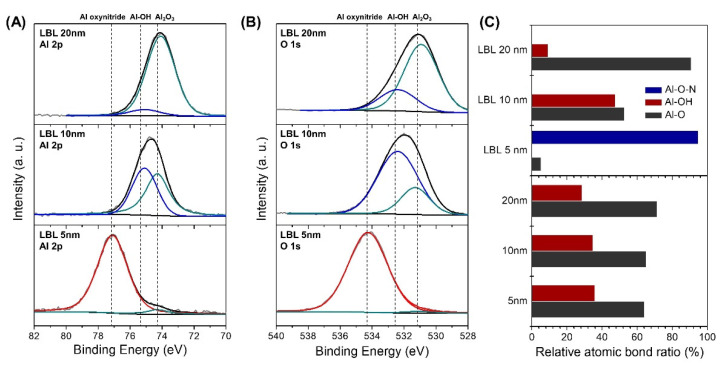
(**A**) Al 2p (**B**) O 1s XPS spectra of LBL Al_2_O_3_ (**C**) relative content of atomic bonding in LBL and bare substrate Al_2_O_3_.

**Figure 3 materials-14-07373-f003:**
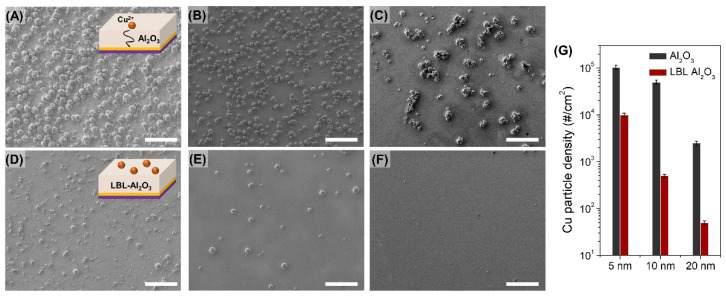
Electro-deposition of Cu particle on barrier bare Al_2_O_3_ (**A**) 5 nm (**B**) 10 nm (**C**) 20 nm and LBL Al_2_O_3_ (**D**) 5nm (**E**) 10 nm (**F**) 20 nm (Scale bar is 10 μm) (**G**) Comparison graph of Cu particle density between bare Al_2_O_3_ and LBL templated Al_2_O_3_ depends on thickness.

**Figure 4 materials-14-07373-f004:**
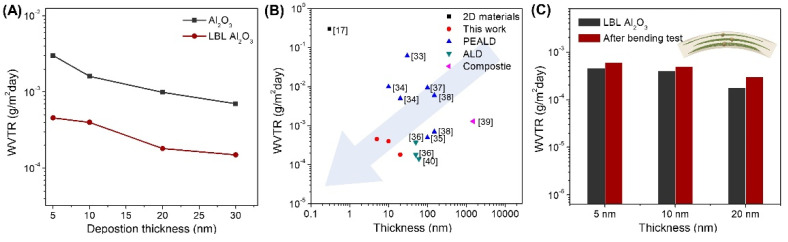
(**A**) WVTR comparison plot of Al_2_O_3_ layers deposited on the bare PEN substrate and hBN/PEI/PEN substrate depending on the thickness (**B**) A plot of WVTR vs. thickness of the PEI/hBN barrier compared to previously reported values (**C**) WVTR values of LBL deposited Al_2_O_3_ layer before and after bending fatigue test (Bending radius: 3 mm (Tensile strain: 4%), 5000 cycles).

## Data Availability

Not applicable.

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
