# Peer review of "hBN Flake Embedded Al2O3 Thin Film for Flexible Moisture Barrier"

_materials, 2021, doi:10.3390/ma14237373_

Round 1

Reviewer 1 Report

The authors has described the introduction of hBN flake embedded Al2O3 film prior to moisture barrier. It may be interesting to the readers of MDPI Materials, however, it is still unclear why hBN is introduced in this work, what is the advantage. The introduction should be revised to explain this more in details.

Figure 1 is too small to read and unclear. Please indicate what the schema want to show.

How did the authors measure the thickness of layers?

Why was Cu2+ introduced to study the moisture? It is not clear to the readers.

Which defects are typically introduced in the layers? 

Author Response

Point-by-point responses to the reviewer’s comments on the Manuscript (Materials-1455480)

We sincerely appreciate the effort that the reviewer has taken in reviewing our manuscript. Thanks to the thoughtful comments of the reviewer, we were able to improve the quality of the manuscript. Changes have been carried out according to the comments in the revised version of the manuscript. We hope that our revision adequately addressed the points of the comments.

Comment 1) The authors have described the introduction of hBN flake embedded Al2O3 film prior to moisture barrier. It may be interesting to the readers of MDPI Materials, however, it is still unclear why hBN is introduced in this work, what is the advantage. The introduction should be revised to explain this more in details.

Reply to comment) As the reviewer suggested, we revised the introduction part of the manuscript to describe the merits using hBN for flexible gas barriers, as follows:

Page 2, lines 54-58

As an alternative to the thin-film barriers, two-dimensional (2D) materials such as graphene and hexagonal boron nitride (hBN) have been studied as flexible gas barriers because of their excellent gas-impermeability, chemical stability, and mechanical flexibility [11, 12]. In particular, hBN has great potential as a material for flexible gas barrier films because of its high optical transparency.

Comment 2) Figure 1 is too small to read and unclear. Please indicate what the schema want to show.

Reply to comment) In order to show the fabrication process and the results more clearly, we modified the scheme and replaced the SEM images with another SEM and TEM images in figure 1 as follows:

Figure 1. hBN templated Al2O3 deposition process. (A) Scheme illustrating initial state of Al2O3 deposition in hBN/PEI template layer, (B) A cross-sectional SEM image of Al2O3/hBN/PEI composite layer after 50 cycles ALD deposition in hBN/PEI template layer. The dashed line in (B) indicates the interface between the PEN substrate and the composite layer. (C) Scheme illustrating saturation state of Al2O3 deposition in hBN/PEI template layer after 200 cycles ALD deposition in hBN/PEI template layer, (D) Cross-sectional TEM images of Al2O3/hBN/PEI composite layer. The inset image is a high-resolution TEM image showing hBN flakes (dashed yellow box) in the composite layer. Scale bars are 100 nm.

Figure S3. (A) A cross-sectional SEM image of hBN/PEI composite layer. (B) A cross-sectional SEM image showing the intermediate state of Al2O3 deposition in hBN/PEI template layer after 100 cycles of ALD deposition in hBN/PEI template layer. The dashed line indicates the interface between the PEN substrate and the composite layer. Scale bars are 100 nm.

Comment 3) How did the authors measure the thickness of layers?

Reply to comment) When Al2O3 is deposited on a bare PEN substrate by the ALD process, a uniform Al2O3 layer is obtained with a thickness of 1 nm per 10 ALD cycles. When Al2O3 is deposited on the hBN/PEI template, however, Al2O3 is nucleated in the voids of the hBN/PEI template and fills the void before being deposited as a uniform layer. Although it is difficult to accurately measure the thickness, the thickness of the hBN/PEI templated Al2O3 layers both at the initial state (50 cycles of ALD deposition) and the intermediate state (100 cycles of ALD deposition) of Al2O3 deposition on the hBN/PEI template were estimated to be about 5-8 nm by measuring the thickness at more than 20 different points which are randomly selected from 5 different SEM images. At the initial and intermediate states, the thickness of the composite layer does not appear to change because Al2O3 mostly nucleates at the edges of the hBN flakes and then deposits in the free voids between the hBN flakes. At the saturation state of Al2O3 deposition on hBN/PEI (200 ALD cycles), the Al2O3 is further deposited on the top surface of the composite after filling the free voids between the hBN flakes, resulting 20 nm thick composite layer. The thickness of the composited layer at the saturation state was measured with cross-sectional TEM images. To address the reviewer’s comment, the manuscript was revised follow as:

Page 3, lines 147-162

Figure 1 shows the deposition process of Al2O3 to form hBN embedded Al2O3 structure. On a bare PEN substrate, a uniform Al2O3 layer is obtained with a thickness of 1 nm per 10 ALD cycles. When Al2O3 is deposited on the hBN/PEI template, however, Al2O3 is nucleated in the voids of the hBN/PEI template and fills the void before being deposited as a uniform layer (Figure 1-(A)). The thickness of the hBN/PEI templated Al2O3 layers both at the initial state (50 cycles of ALD deposition) and the intermediate state (100 cycles of ALD deposition) of Al2O3 deposition on the hBN/PEI template were estimated to be about 5-8 nm by measuring the thickness at more than 20 different points which are randomly selected from 5 different SEM images. At the initial (Figure 1-(B)) and intermediate states (Figure S3-(B)), the thickness of the composite layer does not appear to change because Al2O3 mostly nucleates at the edges of the hBN flakes and then deposits in the free voids between the hBN flakes. At the saturation state of Al2O3 deposition on hBN/PEI (200 ALD cycles), the Al2O3 is further deposited on the top surface of the composite after filling the free voids between the hBN flakes (Figure 1-(C)), resulting in the formation of 20 nm thick composite layer. The thickness of the composited layer at the saturation state was measured with cross-sectional TEM images (Figure 1-(D)).

Comment 4) Why was Cu2+ introduced to study the moisture? It is not clear to the readers.

Reply to comment) If there is no diffusion path of Cu2+ ions in the Al2O3/hBN composite layer, Cu particles are unable to be electrochemically deposited on the composite layer below the breakdown voltage. Since the diameter of Cu2+ ion (0.73 Å) is smaller than that of the water (2.75 Å) or oxygen (3.46 Å) molecules, the absence of diffusion path of Cu2+ ion supports excellent gas barrier performance of the composite. To clearly describe this point, the manuscript has been revised as follows:

Page 5, lines 200-202

Since the diameter of Cu2+ ion (0.73 Å) is smaller than that of the water molecule (2.75 Å), the absence of diffusion path of Cu2+ ion indicates excellent gas barrier performance of the composite.

Comment 5) Which defects are typically introduced in the layers?

Reply to comment) Defects in the hBN/Al2O3 composite layer, causing penetrating water molecules, are divided into two major categories: physical defects, such as pinholes or cracks, and chemical defects formed by un-reacted radicals, low deposition temperature, and the initial state of the ALD deposition process. When the Al2O3 is deposited, Al-O bonds are not completely formed at the initial state, and Al-OH bonds are formed on the surface and inside of the layer. In this work, Cu electroplating and XPS were used together to analyze these defects. To indicate the type of defect, a sentence has been added to the revised manuscript as follows:

Page 1, lines 41-44

The defects in the barrier films, causing penetrating water molecules, are divided into two major types: physical defects, such as pinholes or cracks, and chemical defects formed by un-reacted radicals, low deposition temperature, and the initial state of the ALD deposition process.

Reviewer 2 Report

The manuscript proposed a new method for flexible moisture barrier by utilizing a hexagonal boron nitride embedded Al2O3 thin film. It was claimed that the composite film exhibits both low water vapor transmission ratio and high mechanical stability and hence can be used as a moisture barrier for organic semiconductors. Overall, the method presented in this work is novel, and the topic should be of interest to many readers. Although the presentation is clear and well organized, the authors could improve the quality of the manuscript by providing more details and taking more effort to describe the underlying mechanisms. A few comments are provided below for consideration.

(1) For the benefit of readers/other researchers as well as the reproducibility of the work, the authors should consider providing more details of sample preparation in Sections 2.1 and 2.2. For instance, in Line 85 and 86, the authors only mentioned that the mixture was dialyzed to remove excess urea but provided no details on how dialysis was performed; in Line 96 and 97, the reviewer would expect more information on the PEALD process such as dosage of precursors, the number of cycles, and type of plasma source.

(2) In Line 90, the authors mentioned that negative charges on the substrate were formed by O2 plasma treatment. It is not clear to the reviewer the effect of negative charges. Please explain.

(3) The authors proposed a conversion factor of 240 in Line 115. If the authors referred it to the acceleration factor to convert WVTR at 85/85 to that at another temperature/humidity condition, please articulate it clearly and how the factor is determined.

(4) The 3 cm bending radius described in Line 116 is meaningless unless the authors provide information of their sample substrate and coating thickness/Young’s modulus. It is better that the authors calculate the corresponding thin film strain at the specified bending radius.

(5) It is hard to understand what the authors plan to present in Figure 1 without the readers looking through the main text. In particular, the reviewer had a hard time interpreting the scheme in Fig. 1(b) as ‘the free volume between the flaks being partially filled with Al2O3’ whereas Fig. 1(c) as ‘the free volume being completely filled’. Besides, there is not a description of the cross-section SEM images below the schematic diagrams. It is difficult to recognize the substrate, hBN flaks, and Al2O3 from the images.

(6) There are two Figure S1 (B) in Supplementary Materials. Please correct them.

(7) Reference [30] was not properly cited in Line 152. The defects were recognized as localized bumps visible in AFM while the defects were recognized as Al-OH bonding in this work. There is not a clear connection between these two, despite that both were considered as defects. Indeed, the authors should have explained more the underlying assumption of Al-OH bonding being recognized as defects in Al2O3 thin films.

(8) In Line 156-157, the authors ascribed the Al-OH bond being reduced for the 20 nm Al2O3 deposited in the LBL template to the formation of aluminum oxynitride in the early nucleation stage. It is not clear to the reviewer as aluminum oxynitride only appears in the early state, i.e. ~5nm thick Al2O3, but disappears completely in the intermediate state, i.e. ~10nm thick Al2O3, according to Figure 2(c). It doesn’t look like aluminum oxynitride keeps forming after the early state and inhibits the formation of Al-OH. Please explain.

(9) The reviewer is also curious how the LBL template contributes to the further reduction of WVTR after the Al2O3 thickness larger than 20nm. It can be imagined that the contribution from the LBL template has been saturated at Al2O3 thickness about 20 nm considering the thin nature of the hBN layer, and the further reduction of WVTR for even thicker films should come solely from the defects in the bulk Al2O3 itself. Comments or additional data are welcome.

(10) In Line 208-210, the authors claimed that the increase of WVTR value after the bending test is much less for the composite compared to that for the bare substrate and attributed it to the mechanical stress being released due to PEI. However, in the abstract and Line 76, the authors also mentioned that hBN flaks inhibited crack propagation in barrier film. To the reviewers, these are two different explanations, and none was supported by any experimental or numerical evidence. Please comment and justify the explanation.

Author Response

Point-by-point responses to the reviewer’s comments on the Manuscript (Materials-1455480)

We sincerely appreciate the effort that the reviewer has taken in reviewing our manuscript. Thanks to the thoughtful comments of the reviewer, we were able to improve the quality of the manuscript. Changes have been carried out according to the comments in the revised version of the manuscript. We hope that our revision adequately addressed the points of the comments.

Comment 1) For the benefit of readers/other researchers as well as the reproducibility of the work, the authors should consider providing more details of sample preparation in Sections 2.1 and 2.2. For instance, in Line 85 and 86, the authors only mentioned that the mixture was dialyzed to remove excess urea but provided no details on how dialysis was performed; in Line 96 and 97, the reviewer would expect more information on the PEALD process such as dosage of precursors, the number of cycles, and type of plasma source.

Reply to comment) As reviewer comment. So, more detailed information of experimental methods was added to sections 2.1 and 2.2 as follows:

Page 2, lines 86-108

2.1 The hBN flake exfoliation

The hBN powder (Sigma Aldrich) and urea (98%, Daejung Chemicals) were mixed in a steel container at a weight ratio of 1: 100 (hBN 50 mg, urea 5 g), and the mixture was milled with a rotational speed of 700 rpm for 20 h in Ar atmosphere using a planetary ball mill (Pulverisette 7, Fritsch). To remove excess urea and disperse it in water at a concentration of 1 mg/ml, a mixed powder of urea and hBN flakes was dissolved in 50 ml of deionized water and dialyzed for 1 week using a dialysis kit (Sigma Aldrich).

2.2 Fabrication of LBL template & Al2O3 composite

First, a polyethylene naphthalate (PEN) substrate was treated with O2 plasma for 10 min under 100 W and 100 sccm of O2 gas flow (30 torr) to induce negative charges on the substrate by functionalizing -OH or -O groups on the surface [23]. After that, to adsorb positively charged PEI polymer (MW: 25,000, Sigma Aldrich) by electrostatic interaction, the negatively charged PEN substrate was dipped in a 2 wt % aqueous solution of PEI (pH 10, zeta potential of 59.3 mV) for 10 min and then rinsed with deionized water (DI) for 30s. The PEI-coated substrate was dipped in the aqueous dispersion of hBN flakes (1 mg/mL) for 10 min to adsorb the hBN flake, which has a zeta potential of -39.8 mV, and the resulting substrate was rinsed with DI for 30 s. The zeta potentials of the solutions of PEI polymer and hBN flakes are shown in Figure S1. Then, Al2O3 was deposited on the hBN/PEI substrate by spatial ALD. Al2O3 was deposited using trimethylaluminum (TMA, Sigma Aldrich) as the aluminum precursor and N2O as an oxygen radical source at 80 °C with a remoted plasma power of 150 W. TMA flowed with Ar gas as a carrier gas in 50 sccm and N2O gas in 20 sccm. To control the layer thickness, about 10 cycles per 1 nm thickness were repeated, and the moving speed of the spatial ALD was 125 mm/s.

Comment 2) In Line 90, the authors mentioned that negative charges on the substrate were formed by O2 plasma treatment. It is not clear to the reviewer the effect of negative charges. Please explain.

Reply to comment) By O2 plasma treatment of the PEN substrate, the surface of the substrate is oxidized to form -OH or -O groups on the surface, which results in the formation of negative charges on the substrate [Radiation Physics and Chemistry 2007, 76, 1011–1016]. Since the amine functional groups of the PEI polymer are readily converted to positively charged ammonium units in an aqueous solution, the PEI polymer can be adsorbed onto the negatively charged PEN substrate by electrostatic interaction. As shown in the figure below, the aqueous solution of PEI polymer has pH 10 and zeta potential of 59.3 mV, indicating the PEI polymers are positively charged by protonation of amine groups in PEI. On the PEI absorbed substrate, PEI-hBN composite layer is formed by adsorbing negatively charged hBN flakes with the zeta potential of -39.8 mV.

Figure S1. The zeta potential graph of the solutions of PEI polymer and hBN flakes.

Comment 3) The authors proposed a conversion factor of 240 in Line 115. If the authors referred it to the acceleration factor to convert WVTR at 85/85 to that at another temperature/humidity condition, please articulate it clearly and how the factor is determined.

Reply to comment) The acceleration factor can be found in our previous work [J. Korean Phys. Soc. 2018, 73, 45-52]. It was determined by comparing the values measured at 25oC/50% RH (25oC and relative humidity of 50%) with those measured at 85oC/85% RH. To describe this part, the experimental section was revised as follow:

Page 3, lines 128-130

The conversion factor was determined by comparing the WVTR values measured at 25oC/50% RH (25oC and relative humidity of 50%) with those measured at 85oC/85% RH in our previous work [J. Korean Phys. Soc. 2018, 73, 45-52].

Comment 4) The 3 cm bending radius described in Line 116 is meaningless unless the authors provide information of their sample substrate and coating thickness/Young’s modulus. It is better that the authors calculate the corresponding thin film strain at the specified bending radius.

Reply to comment) We thank the reviewer for the valuable comment. To address the comment, the strain of the barrier layer is calculated using equation [Nanoscale Adv. 2019, 1, 1215], as follows:

where t and r are the thickness of film and bending radius. Since we conducted a bending test with a bending radius of 3 mm and a 200 μm PEN substrate, 3 mm bending radius corresponds to 4 % strain by the conversion equation. Both bending radius and corresponding strain were specified in the revised manuscript as follow:

Page 6, lines 247-249

In addition, the flexibility was confirmed by measuring the WVTR value after 5000 cycles of bending fatigue test at a bending radius of 3 mm, which corresponds to 4 % stain applied (Figure 4-(C)).

Figure 4. (A) WVTR comparison plot of Al2O3 layers deposited on the bare PEN substrate and hBN/PEI/PEN substrate depending on the thickness (B) A plot of WVTR vs. thickness of the PEI/hBN barrier compared to previously reported values (C) WVTR values of LBL deposited Al2O3 layer before and after bending fatigue test (Bending radius: 3 mm (Tensile strain: 4%), 5000 cycles).

Comment 5) It is hard to understand what the authors plan to present in Figure 1 without readers looking through the main text. In particular, the reviewer had a hard time interpreting the scheme in Fig. 1(b) as ‘the free volume between the flaks being partially filled with Al2O3’ whereas Fig. 1(c) as ‘the free volume being completely filled’. Besides, there is not a description of the cross-section SEM images below the schematic diagrams. It is difficult to recognize the substrate, hBN flaks, and Al2O3 from the images.

Reply to comment) In order to show the fabrication process and the results more clearly, we modified the scheme and replaced the SEM images with another SEM and TEM images in figure 1 as follows:

Figure 1. hBN templated Al2O3 deposition process. (A) Scheme illustrating initial state of Al2O3 deposition in hBN/PEI template layer, (B) A cross-sectional SEM image of Al2O3/hBN/PEI composite layer after 50 cycles ALD deposition in hBN/PEI template layer. The dashed line in (B) indicates the interface between the PEN substrate and the composite layer. (C) Scheme illustrating saturation state of Al2O3 deposition in hBN/PEI template layer after 200 cycles ALD deposition in hBN/PEI template layer, (D) Cross-sectional TEM images of Al2O3/hBN/PEI composite layer. The inset image is a high-resolution TEM image showing hBN flakes (dashed yellow box) in the composite layer. Scale bars are 100 nm.

Figure S3. (A) A cross-sectional SEM image of hBN/PEI composite layer. (B) A cross-sectional SEM image showing the intermediate state of Al2O3 deposition in hBN/PEI template layer after 100 cycles of ALD deposition in hBN/PEI template layer. The dashed line indicates the interface between the PEN substrate and the composite layer. Scale bars are 100 nm.

To address the reviewer’s comment, the manuscript was also revised follow as:

Page 3, lines 147-162

Figure 1 shows the deposition process of Al2O3 to form hBN embedded Al2O3 structure. On a bare PEN substrate, a uniform Al2O3 layer is obtained with a thickness of 1 nm per 10 ALD cycles. When Al2O3 is deposited on the hBN/PEI template, however, Al2O3 is nucleated in the voids of the hBN/PEI template and fills the void before being deposited as a uniform layer (Figure 1-(A)). The thickness of the hBN/PEI templated Al2O3 layers both at the initial state (50 cycles of ALD deposition) and the intermediate state (100 cycles of ALD deposition) of Al2O3 deposition on the hBN/PEI template were estimated to be about 5-8 nm by measuring the thickness at more than 20 different points which are randomly selected from 5 different SEM images. At the initial (Figure 1-(B)) and intermediate states (Figure S3-(B)), the thickness of the composite layer does not appear to change because Al2O3 mostly nucleates at the edges of the hBN flakes and then deposits in the free voids between the hBN flakes. At the saturation state of Al2O3 deposition on hBN/PEI (200 ALD cycles), the Al2O3 is further deposited on the top surface of the composite after filling the free voids between the hBN flakes (Figure 1-(C)), resulting in the formation of 20 nm thick composite layer. The thickness of the composited layer at the saturation state was measured with cross-sectional TEM images (Figure 1-(D)).

Comment 6) There are two Figure S2 (B) in Supplementary Materials. Please correct them

Reply to comment) We thank the reviewer for pointing out the mistake. The caption of Figure S2 was revised as follows:

Figure S2. (A) AFM image and height profile of hBN flakes. (B) B 1s (C) N 1s XPS spectra of exfoliated hBN flake

Comment 7) Reference [30] was not properly cited in Line 152. The defects were recognized as localized bumps visible in AFM while the defects were recognized as Al-OH bonding in this work. There is not a clear connection between these two, despite that both were considered as defects. Indeed, the authors should have explained more the underlying assumption of Al-OH bonding being recognized as defects in Al2O3 thin films.

Reply to comment) Al-OH bonding is formed because the precursor does not fully react during the Al2O3 deposition process. Due to the incomplete reaction, Al-OH is observed when Al2O3 is depositied with under 5 nm thickness, or at a low temperature. The Al-OH bonds act as point defects of Al2O3 layer or diffusion path of moisture. In a previous report [Ceramics International 2019, 45, 19105–19112], Al2O3 was deposited at various oxygen radical sources and deposition temperatures, and at low deposition temperatures, moisture was penetrated through Al-OH bonds. The sentence at Line 152 of the original manuscript was revised, and the previous report mentioned above was added as a reference, as follows:

Page 4, lines 183-185

In the saturation state of depositing Al2O3 with the thickness of >10 nm, Al-OH bonds are converted to Al-O bonds due to reaction with oxygen radicals as the number of deposition cycles increases [30, Ceramics International 2019, 45, 19105–19112].

Comment 8) In Line 156-157, the authors ascribed the Al-OH bond being reduced for the 20 nm Al2O3 deposited in the LBL template to the formation of aluminum oxynitride in the early nucleation stage. It is not clear to the reviewer as aluminum oxynitride only appears in the early state, i.e. ~5nm thick Al2O3, but disappears completely in the intermediate state, i.e. ~10 nm thick Al2O3, according to Figure 2(c). It doesn’t look like aluminum oxynitride keeps forming after the early state and inhibits the formation of Al-OH. Please explain.

Reply to comment) We used angled resolved XPS to view functional groups on the surface. Although the accurate penetration depth of X-ray during the analysis is unknown, a signal at the top surface with less than 10 nm thickness was observed. For this reason, aluminum oxynitride was observed in the initial state (50 deposition cycles) during the ALD process, but it is confirmed that Al-OH on the surface was observed by additional deposition (100~200 deposition cycles). In the initial state, Al-OH, which has not been converted to Al-O bond, is suppressed by binding to the amine of hBN flakes and forming aluminum oxynitride. After all the available amine groups are used (i.e., after the initial stage), the formation of Al-OH (exposed to the Al2O3 surface) cannot be inhibited.

For more accurate XPS analysis depending on thickness, we tried to analyze XPS using Ar sputter, but the barrier layer was not etched. To describe the XPS analysis in more detail, the manuscript was revised as follows:

Page 3, lines 114-117

Functional groups and defects in LBL-Al2O3 and Al2O3 were evaluated using X-ray photoelectron spectroscopy (XPS, ESCALAB 250Xi) with an angle-resolved mode at 60o tilting. Although the accurate penetration depth of X-ray during the analysis is unknown, a signal at the top surface with less than 10 nm thickness was observed [Journal of Electron Spectroscopy and Related Phenomena 1995, 73, 25-52].

Comment 9) The reviewer is also curious how the LBL template contributes to the further reduction of WVTR after the Al2O3 thickness larger than 20nm. It can be imagined that the contribution from the LBL template has been saturated at Al2O3 thickness about 20 nm considering the thin nature of the hBN layer, and the further reduction of WVTR for even thicker films should come solely from the defects in the bulk Al2O3 itself. Comments or additional data are welcome.

Reply to comment) We also measured the 30 nm thick Al2O3 deposited on bare PEN, and hBN/PEI templated substrates. As the reviewer mentioned, when 10 nm of Al2O3 was additionally deposited after saturation, the WVTR value was only slightly reduced from 1.8×10-4 to 1.5×10-4 g/m2 day due to additional Al2O3. To address the reviewer’s comment, the manuscript was revised follow as:

Page 6, lines 234-235

When 10 nm thick Al2O3 was additionally deposited after the saturation, the WVTR value was only slightly reduced to 1.5×10-4 g/m2 day.

Figure 4. (A) WVTR comparison plot of Al2O3 layers deposited on the bare PEN substrate and hBN/PEI/PEN substrate depending on the thickness (B) A plot of WVTR vs. thickness of the PEI/hBN barrier compared to previously reported values (C) WVTR values of LBL deposited Al2O3 layer before and after bending fatigue test (Bending radius: 3 mm (Tensile strain: 4%), 5000 cycles).

Figure S7. Representative normalized conductance vs. time for Al2O3 layers deposited on various substrates and various thicknesses.

Comment 10) In Line 208-210, the authors claimed that the increase of WVTR value after the bending test is much less for the composite compared to that for the bare substrate and attributed it to the mechanical stress being released due to PEI. However, in the abstract and Line 76, the authors also mentioned that hBN flaks inhibited crack propagation in barrier film. To the reviewers, these are two different explanations, and none was supported by any experimental or numerical evidence. Please comment and justify the explanation.

Reply to comment) When hBN/polymer composite is fabricated, stress relaxation occurs depending on the ratio and characteristics of the polymer [Polymer 2020, 208, 122964]. For hBN/inorganic composite materials, crack propagation was restricted during a mechanical elongation test [Metals and Materials International 2021, 27,802–814].

Page 6, lines 250-254

As illustrated in the inset of Figure 4-(C), the mechanical stress can be released due to the PEI polymer layer, and crack propagation can also be restricted by hBN flakes in the composite [Polymer 2020, 208, 122964], [Metals and Materials International 2021, 27, 802–814]. For Al2O3 layers deposited on bare PEN substrate, however, WVTR values increased by more than 200% after the bending test (Figure S8).

Reviewer 3 Report

Dear Authors, 

In my opinion your work interesting and relatively well presented. I have some some comments on your research:

  1. there are few mistakes in the text, e.g. l. 17 double "the" - check carefully the rest of the text. 
  2. units for density should be cm-2 , not /cm2
  3. descriptions in Figures are very small - I would suggest to increase it, maybe even Figures for better reading and understanding
  4. The  thickness was measured by a field-effect scanning electron microscopy - what was the accuracy of the measurement? What was the deviation? 
  5. Density - the same comment as to the thickness: measurement, manually or automatically with the software for the image analysis, deviation of the distribution.
  6. Further research should be indicated in the Conclusions. 

Regards, 

Reviewer

Author Response

Point-by-point responses to the reviewer’s comments on the Manuscript (Materials-1455480)

We sincerely appreciate the effort that the reviewer has taken in reviewing our manuscript. Thanks to the thoughtful comments of the reviewer, we were able to improve the quality of the manuscript. Changes have been carried out according to the comments in the revised version of the manuscript. We hope that our revision adequately addressed the points of the comments.

Comment 1) there are few mistakes in the text, e.g. l. 17 double "the" - check carefully the rest of the text. units for density should be cm-2 , not /cm2

Reply to comment) We thank the reviewer for the valuable comment. The mistakes are corrected in the revised manuscript as follow:

Page 1, lines 16-18

Because the hBN flakes in Al2O3 thin film increase the diffusion path of moisture, the composite layer has a low water vapor transmission ratio (WVTR) value of 1.8 × 10-4 g/m2 day.

Page 5, lines 205-212

In 5 nm thick Al2O3 deposited on the bare substrate, Cu2+ ions rapidly penetrated through cracks or point defects of the Al2O3 layer and nucleated to form Cu particles in high densi-ty of ~105 cm-2 (Figure 3-(A) ~ (C)). As the thickness increased to 20 nm, the Cu particle density decreased due to the reduction of point defects, but the density was still ~103 cm-2. However, when Al2O3 is deposited on the hBN/PEI template, the Cu particle density is significantly lower than the value of Al2O3 on a bare PEN substrate (Figure 3-(D) ~ (F)). Especially, when 20 nm thick Al2O3 was deposited on hBN/PEI template, the particle density was minimized to less than 50 cm-2.

Comment 2) descriptions in Figures are very small - I would suggest to increase it, maybe even Figures for better reading and understanding

Reply to comment) In order to show the fabrication process and the results more clearly, we modified the scheme and replaced the SEM images with another SEM and TEM images in figure 1 as follows:

Figure 1. hBN templated Al2O3 deposition process. (A) Scheme illustrating initial state of Al2O3 deposition in hBN/PEI template layer, (B) A cross-sectional SEM image of Al2O3/hBN/PEI composite layer after 50 cycles ALD deposition in hBN/PEI template layer. The dashed line in (B) indicates the interface between the PEN substrate and the composite layer. (C) Scheme illustrating saturation state of Al2O3 deposition in hBN/PEI template layer after 200 cycles ALD deposition in hBN/PEI template layer, (D) Cross-sectional TEM images of Al2O3/hBN/PEI composite layer. The inset image is a high-resolution TEM image showing hBN flakes (dashed yellow box) in the composite layer. Scale bars are 100 nm.

Figure S3. (A) A cross-sectional SEM image of hBN/PEI composite layer. (B) A cross-sectional SEM image showing the intermediate state of Al2O3 deposition in hBN/PEI template layer after 100 cycles of ALD deposition in hBN/PEI template layer. The dashed line indicates the interface between the PEN substrate and the composite layer. Scale bars are 100 nm.

Comment 3) The thickness was measured by a field-effect scanning electron microscopy - what was the accuracy of the measurement? What was the deviation?

Reply to comment) When Al2O3 is deposited on a bare PEN substrate by the ALD process, a uniform Al2O3 layer is obtained with a thickness of 1 nm per 10 ALD cycles. When Al2O3 is deposited on the hBN/PEI template, however, Al2O3 is nucleated in the voids of the hBN/PEI template and fills the void before being deposited as a uniform layer. Although it is difficult to accurately measure the thickness, the thickness of the hBN/PEI templated Al2O3 layers both at the initial state (50 cycles of ALD deposition) and the intermediate state (100 cycles of ALD deposition) of Al2O3 deposition on the hBN/PEI template were estimated to be about 5-8 nm by measuring the thickness at more than 20 different points which are randomly selected from 5 different SEM images. At the initial and intermediate states, the thickness of the composite layer does not appear to change because Al2O3 mostly nucleates at the edges of the hBN flakes and then deposits in the free voids between the hBN flakes. At the saturation state of Al2O3 deposition on hBN/PEI (200 ALD cycles), the Al2O3 is further deposited on the top surface of the composite after filling the free voids between the hBN flakes, resulting 20 nm thick composite layer. The thickness of the composited layer at the saturation state was measured with cross-sectional TEM images. To address the reviewer’s comment, the manuscript was revised follow as:

Page 3, lines 147-162

Figure 1 shows the deposition process of Al2O3 to form hBN embedded Al2O3 structure. On a bare PEN substrate, a uniform Al2O3 layer is obtained with a thickness of 1 nm per 10 ALD cycles. When Al2O3 is deposited on the hBN/PEI template, however, Al2O3 is nucleated in the voids of the hBN/PEI template and fills the void before being deposited as a uniform layer (Figure 1-(A)). The thickness of the hBN/PEI templated Al2O3 layers both at the initial state (50 cycles of ALD deposition) and the intermediate state (100 cycles of ALD deposition) of Al2O3 deposition on the hBN/PEI template were estimated to be about 5-8 nm by measuring the thickness at more than 20 different points which are randomly selected from 5 different SEM images. At the initial (Figure 1-(B)) and intermediate states (Figure S3-(B)), the thickness of the composite layer does not appear to change because Al2O3 mostly nucleates at the edges of the hBN flakes and then deposits in the free voids between the hBN flakes. At the saturation state of Al2O3 deposition on hBN/PEI (200 ALD cycles), the Al2O3 is further deposited on the top surface of the composite after filling the free voids between the hBN flakes (Figure 1-(C)), resulting in the formation of 20 nm thick composite layer. The thickness of the composited layer at the saturation state was measured with cross-sectional TEM images.

Comment 4) Density - the same comment as to the thickness: measurement, manually or automatically with the software for the image analysis, deviation of the distribution

Reply to comment) The density of Cu particles was manually measured. After Cu electro-plating in an area of about 0.3 x 0.3 cm2, the density was measured by counting the number of Cu particles per area using SEM images (with the size of 50 x 40 mm2) taken at more than 20 different locations of 0.3 x 0.3 cm2 area. The deviation of the density of copper particles was about 10%. The figure 3 was revised by adding the deviation as an error bar as follows:

Figure 3. Electro-deposition of Cu particle on barrier bare Al2O3 (A) 5 nm (B) 10 nm (C) 20 nm and LBL Al2O3 (D) 5nm (E) 10 nm (F) 20 nm (Scale bar is 10 μm) (G) Comparison graph of Cu particle density between bare Al2O3 and LBL templated Al2O3 depends on thickness.

Comment 5) Further research should be indicated in the Conclusions.

Reply to comment) As the reviewer suggested, a couple of sentences describing the potential and required further works were added in the conclusion part of the revised manuscript, as follows:

Page 7, lines 268-271

In this study, a WVTR value of about 10-4 g/m2 day was obtained even with a single layer with a thin thickness through the hBN embedded Al2O3 structure. To apply this approach to OLED applications, it is necessary to further research for WVTR improvement through multi-layer staking.

Reviewer 4 Report

The manuscript is interesting, but has some drawbacks that need to be corrected.

The quality of photographs in figure 1 is very poor. The real microstructure was not presented in the manuscript. The real thickness of the films and the distribution of the boron nitride flakes were not presented clearly enough. Thickness measurements are crucial for this work, so the values should be reliable. Thickness measurements are crucial for this work and therefore the values should be reliable.

The higher SEM magnification, maybe combined with other cross-section specimen preparation protocol (e.g. cross-section ion milling) should be used. If SEM technique is not sufficient, TEM (transmission electron microscopy) should be applied to show the microstructure of the layers.

Author Response

Point-by-point responses to the reviewer’s comments on the Manuscript (Materials-1455480)

We sincerely appreciate the effort that the reviewer has taken in reviewing our manuscript. Thanks to the thoughtful comments of the reviewer, we were able to improve the quality of the manuscript. Changes have been carried out according to the comments in the revised version of the manuscript. We hope that our revision adequately addressed the points of the comments.

Comment 1) The quality of photographs in figure 1 is very poor. The real microstructure was not presented in the manuscript. The real thickness of the films and the distribution of the boron nitride flakes were not presented clearly enough. Thickness measurements are crucial for this work, so the values should be reliable. Thickness measurements are crucial for this work and therefore the values should be reliable. The higher SEM magnification, maybe combined with other cross-section specimen preparation protocol (e.g. cross-section ion milling) should be used. If SEM technique is not sufficient, TEM (transmission electron microscopy) should be applied to show the microstructure of the layers.

Reply to comment) In order to show the fabrication process and the results more clearly, we modified the scheme and replaced the SEM images with another SEM and TEM images in figure 1 as follows:

Figure 1. hBN templated Al2O3 deposition process. (A) Scheme illustrating initial state of Al2O3 deposition in hBN/PEI template layer, (B) A cross-sectional SEM image of Al2O3/hBN/PEI composite layer after 50 cycles ALD deposition in hBN/PEI template layer. The dashed line in (B) indicates the interface between the PEN substrate and the composite layer. (C) Scheme illustrating saturation state of Al2O3 deposition in hBN/PEI template layer after 200 cycles ALD deposition in hBN/PEI template layer, (D) Cross-sectional TEM images of Al2O3/hBN/PEI composite layer. The inset image is a high-resolution TEM image showing hBN flakes (dashed yellow box) in the composite layer. Scale bars are 100 nm.

Figure S3. (A) A cross-sectional SEM image of hBN/PEI composite layer. (B) A cross-sectional SEM image showing the intermediate state of Al2O3 deposition in hBN/PEI template layer after 100 cycles of ALD deposition in hBN/PEI template layer. The dashed line indicates the interface between the PEN substrate and the composite layer. Scale bars are 100 nm.

When Al2O3 is deposited on a bare PEN substrate by the ALD process, a uniform Al2O3 layer is obtained with a thickness of 1 nm per 10 ALD cycles. When Al2O3 is deposited on the hBN/PEI template, however, Al2O3 is nucleated in the voids of the hBN/PEI template and fills the void before being deposited as a uniform layer. Although it is difficult to accurately measure the thickness, the thickness of the hBN/PEI templated Al2O3 layers both at the initial state (50 cycles of ALD deposition) and the intermediate state (100 cycles of ALD deposition) of Al2O3 deposition on the hBN/PEI template were estimated to be about 5-8 nm by measuring the thickness at more than 20 different points which are randomly selected from 5 different SEM images. At the initial and intermediate states, the thickness of the composite layer does not appear to change because Al2O3 mostly nucleates at the edges of the hBN flakes and then deposits in the free voids between the hBN flakes. At the saturation state of Al2O3 deposition on hBN/PEI (200 ALD cycles), the Al2O3 is further deposited on the top surface of the composite after filling the free voids between the hBN flakes, resulting 20 nm thick composite layer. The thickness of the composited layer at the saturation state was measured with cross-sectional TEM images. To address the reviewer’s comment, the manuscript was revised follow as:

Page 3, lines 147-162

Figure 1 shows the deposition process of Al2O3 to form hBN embedded Al2O3 structure. On a bare PEN substrate, a uniform Al2O3 layer is obtained with a thickness of 1 nm per 10 ALD cycles. When Al2O3 is deposited on the hBN/PEI template, however, Al2O3 is nucleated in the voids of the hBN/PEI template and fills the void before being deposited as a uniform layer (Figure 1-(A)). The thickness of the hBN/PEI templated Al2O3 layers both at the initial state (50 cycles of ALD deposition) and the intermediate state (100 cycles of ALD deposition) of Al2O3 deposition on the hBN/PEI template were estimated to be about 5-8 nm by measuring the thickness at more than 20 different points which are randomly selected from 5 different SEM images. At the initial (Figure 1-(B)) and intermediate states (Figure S3-(B)), the thickness of the composite layer does not appear to change because Al2O3 mostly nucleates at the edges of the hBN flakes and then deposits in the free voids between the hBN flakes. At the saturation state of Al2O3 deposition on hBN/PEI (200 ALD cycles), the Al2O3 is further deposited on the top surface of the composite after filling the free voids between the hBN flakes (Figure 1-(C)), resulting in the formation of 20 nm thick composite layer. The thickness of the composited layer at the saturation state was measured with cross-sectional TEM images.

Round 2

Reviewer 1 Report

The authors have addressed on reviewers' comments point-by-point and adjusted the manuscript. It is now suitable for the publication.

Author Response

Point-by-point responses to the reviewer’s comments on the Manuscript (Materials-1455480)

We sincerely appreciate the effort that the editor and reviewers have taken in reviewing our manuscript. Thanks to the thoughtful comments of the reviewers, we were able to improve the quality of the manuscript.

Comment 1) The authors have addressed on reviewers' comments point-by-point and adjusted the manuscript. It is now suitable for the publication.

Reviewer 2 Report

The quality of the manuscript has been improved substantially, and the reviewer's concerns were eased. Thanks for the work. 

Author Response

Point-by-point responses to the reviewer’s comments on the Manuscript (Materials-1455480)

We sincerely appreciate the effort that the editor and reviewers have taken in reviewing our manuscript. Thanks to the thoughtful comments of the reviewers, we were able to improve the quality of the manuscript.

Comment 1) The quality of the manuscript has been improved substantially, and the reviewer's concerns were eased. Thanks for the work.

Reviewer 4 Report

In figure 1, TEM image could be a better quality. SEM image is still unclear, so should be replaced.

Author Response

Point-by-point responses to the reviewer’s comments on the Manuscript (Materials-1455480)

We sincerely appreciate the effort that the editor and reviewers have taken in reviewing our manuscript. Thanks to the thoughtful comments of the reviewers, we were able to improve the quality of the manuscript. Changes have been carried out according to the comments in the revised version of the manuscript. We hope that our revision adequately addressed the points of the comments.

Comment 1) In figure 1, TEM image could be a better quality. SEM image is still unclear, so should be replaced.

Reply to comment) As reviewer’s comment, the SEM image in Figure 1(B) was replaced with another image. We believe that the Al2O3/hBN/PEI composite layer is more clearly visualized in the new image.

Figure 1. hBN templated Al2O3 deposition process (A) Scheme illustrating initial state of Al2O3 deposition in hBN/PEI template layer, (B) A cross-sectional SEM image of Al2O3/hBN/PEI composite layer, where 5 nm thick Al2O3 is deposited in hBN/PEI template layer. Dashed line indicates the interface between PEN substrate and the composite layer. (C) Scheme illustrating saturation state of Al2O3 deposition in hBN/PEI template layer, (D) Cross-sectional TEM images of Al2O3/hBN/PEI composite layer, where 20 nm thick Al2O3 is deposited in hBN/PEI template layer. Inset image is high-resolution TEM image showing hBN flakes (dashed yellow box) in the composite layer. Scale bars are 100 nm.
